# Knowledge, attitude and practice towards antenatal physical exercise among pregnant women in Ethiopia: A systematic review and meta-analysis

**Abebaw Wasie Kasahun**[1]*, **Solomon Shitu**[2], **Berhanu Abebaw Mekonnen**[3], **Michael Hawlet**[4], **Amare Zewdie**[1]

1 Department of Public Health, College of Medicine and Health Science, Wolkite University, Wolkite, Ethiopia, 2 Department of Midwifery, College of Medicine and Health Science, Wolkite University, Wolkite, Ethiopia, 3 Department of Nutrition and Dietetics, College of Medicine and Health Sciences, Bahir Dar University, Bahir Dar, Ethiopia, 4 School of Medicine, College of Medicine and Health sciences, Wolkite University, Wolkite, Ethiopia

* abebawasie@gmail.com

**Data Availability Statement:** All relevant data are within the paper and its Supporting Information files.

## Abstract

### Introduction

Physical activity and exercise during pregnancy have paramount importance for both the mother and fetus. Regardless of the benefits of exercise during pregnancy, significant proportion of women usually opt sedentary lifestyle during pregnancy. The aim of this systematic review and meta-analysis is to synthesis evidences on knowledge, attitude, and practice towards antenatal physical exercise among pregnant women in Ethiopia.

### Methods

The systematic review and meta-analysis was conducted on knowledge, attitude, practice, and associated factors towards antenatal physical exercise among pregnant women in Ethiopia. The systematic review and meta-analysis protocol was registered on PROSPERO website with registration number CRD42023444723. Articles were searched on international databases using medical subject heading and keywords. After ensuring eligibility, data were extracted using Microsoft excel and imported to STATA 17 for analysis. Cochran Q test and $I^2$ statistics were used to check presence of heterogeneity. Weighted Inverse variance random effect model was used to estimate the pooled level of knowledge, attitude, and practice on antenatal physical exercise among pregnant women in Ethiopia. Funnel plot and egger's test were used to check presence of publication bias.

### Results

A total of 11 studies were included in this systematic review and meta-analysis. The pooled prevalence of adequate knowledge, favorable attitude and good practice towards antenatal physical exercise were 46.04% with 95% CI (44.45%-47.63%), 43.71%, 95% CI (41.95%-45.46%) and 34.06, 95% CI (20.04%-48.08%) respectively. Good knowledge AOR 2.38

**Funding:** The authors received no specific funding for this work.

**Competing interests:** The authors have declared that no competing interests exist.

**Abbreviations:** ACOG, American College of Obstetrics and Gynecology; CoCo/PEO, Condition, Context, Population, Exposure, Outcome; EPI, Expanded program on Immunization; Fig, Figure; PRISMA, Preferred Reporting Items for Systematic Reviews and Meta-Analyses; MeSH, Medical subject heading; SNNPR, South Nations, Nationalities and People Region.

95% CI(1.80–3.14), unfavorable attitude AOR 0.43 95% CI (0.32–0.58), having no pre-pregnancy habit of physical exercise AOR 0.44 95 CI (0.24–0.79) and having diploma or above education status AOR 3.39 95% CI (1.92–5.98) were significantly associated with practice of antenatal physical exercise.

## Conclusion

The level of knowledge, attitude and practice towards antenatal physical exercise among pregnant women was far below the recommended level in Ethiopia. Knowledge, attitude, education status and pre-pregnancy habit of physical exercise were significantly associated factors with antenatal physical exercise practice. It is highly essential to disseminate health information on the benefits of antenatal physical exercise for all pregnant women during antenatal care contacts.

## Introduction

Exercise is any physical activity performed in structured, planned, repetitive bodily movement to bring at least one components of physical body fitness which is an essential element of healthy life style [1].

Regular physical activity is essential at all stages of life including during pregnancy; it provides physical, emotional and mental health benefits. According to the American college of Obstetrician and Gynecologists (ACOG), physical activity and exercise during pregnancy are associated with low risks and it has proven beneficial to most women, however, there must be some modifications to the ordinary style of exercising as pregnancy carries normal physiological and anatomical changes as well as for safeguarding fetal requirements. ACOG recommends that women with low-risk pregnancies can safely participate in moderate-intensity physical activities for at least 30 minutes on most or all days of the week. However, a thorough prior clinical examination has to be done before recommending physical exercise to ensure that the client has no medical conditions to deter exercise [2].

The aforementioned recommendation is given based on the research findings that portrayed antenatal physical exercise have reduced the incidence of adverse outcomes including preeclampsia, gestational diabetes, macrosomia, gestational weight gain, low back pain, insomnia, heartburn, nausea and others [3–5]. On the other hand, physical inactivity during pregnancy is associated excessive weight gain, maternal obesity and its related complications including gestational diabetic mellitus [5–7]. Furthermore, physical exercise/activity during pregnancy decreases incidence of cesarean delivery, destructive vaginal delivery, and postpartum recovery time; it also reduces depressive disorders of women during postpartum period [8–11].

Concerns that antenatal physical exercise causes miscarriage, fetal growth restriction, preterm delivery and musculoskeletal injury are refuted by rigorous studies among women with uncomplicated pregnancies [10,12,13]. In the absence of obstetric or medical complications, physical activity is safe and desirable for all pregnant women; hence it is commendable to encourage women to continue exercising or encourage commencing physical exercises.

Regardless of the benefits of exercise during pregnancy, significant proportion of women usually opt sedentary lifestyle during pregnancy; it is due to the fact that physiological and psychological changes in pregnancy promote sedentary behavior among women. Moreover,

women are not getting sate of the art advice on the benefits of antenatal exercises and considerable proportion of them believed that physical exercise is risky to the fetus [14–16]. Thus, it is imperative to find out the level of knowledge, attitude and practices of pregnant women towards antenatal physical exercise so as to institute evidence based interventions for having desirable lifestyle during pregnancy.

In Ethiopia, there is no unequivocal knowledge on what is known, believed and practiced concerning antenatal exercise, besides the enablers and barriers of physical exercise during pregnancy are not well established. Although, there are few studies aimed at unpacking the level of pregnant women's knowledge, attitude and practices towards antenatal physical exercise; the results are too divergent and not nationally representative [17–25]. Therefore, it is crucial to synthesize the available evidences to produce a single estimate of the knowledge, attitude and practices of antenatal physical exercise for better decision making purpose in Ethiopia. In accordance with the aforementioned essence, a systematic review and meta-analysis was conducted on knowledge, attitude and practices of a physical exercise among pregnant women in Ethiopia.

## Methods

### Study design and setting

The PROSPERO website (http://www.library.ucsf.edu/) was checked whether the title was previously done or whether an ongoing systematic review and meta-analysis exists on the title to avoid duplications. Accordingly, there was no registered published and/or ongoing systematic review and meta-analysis on antenatal physical exercise and associated factors among pregnant women in Ethiopia. A systematic review and meta-analysis was conducted to determine the pooled level of knowledge, attitude and practice of antenatal physical exercise and associated factors among pregnant women in Ethiopia. This systematic review and meta-analysis was conducted following the preferred reporting items for systematic review and meta -analysis [26] see supplementary file 2 (S1 Checklist).

### Eligibility criteria

Studies reporting the level of knowledge and/or attitude and/or practice and /or associated factors of antenatal physical exercise among pregnant women in Ethiopia were included in this systematic review and meta-analysis. Published research articles and unpublished researches including preprints and grey literature written in the English language were eligible regardless of the study design and time of publication or time of the study. Articles published at any time until the end of our search (July12, 2023) were included in this systematic review and metaanalysis. Articles that did not report the outcome variables or articles with unrelated outcome variable to the interest of this systematic review and meta-analysis were excluded. Furthermore, articles with no full abstracts, letters, commentaries, editorials, and unspecified reports were not considered for this systematic review and meta-analysis.

### Information sources

Databases including PubMed, Scopus, African Journals Online, Web of Science, and Google Scholar were used to find research articles on knowledge, attitude and practice of antenatal physical exercise among pregnant women in Ethiopia. Furthermore, a hand search was made on references of retrieved articles to find all eligible articles for this systematic review and meta-analysis.

## Search strategies

Firstly preliminary search was done using medical subject headings. Secondly, keywords were developed using key terms from retrieved articles on the preliminary search. Finally, medical subject headings and keywords were used to search articles on medical and health sciences research databases and other search engines. Furthermore, librarians were consulted to find unpublished research works on our area of interest for this systematic review and meta-analysis.

Search terms were designed using CoCo/PEO guidelines [27,28]. Searching for articles was done using medical subject headings (MeSH) and key terms through online databases. Boolean operators including "AND" and "OR" were used to link MeSH and the keywords for searching purposes. Accordingly, the developed search terms consist of "knowledge" OR "attitude" OR "practice" AND "exercise" OR "physical exercise" OR "physical activity" "pregnant women" OR "antenatal" AND "predictors" OR "determinants" OR "factors" AND "Ethiopia" (S1 File).

## Quality assessment and selection process

Modified Newcastle Ottawa quality assessment scale for cross-sectional studies was used to assess the quality of studies. The scale has three categories (selection, comparability and outcome) with a maximum possible score of nine points. Each study was assessed against the criteria set on the scale and the studies that have awarded a score of 7 or above are labeled to have good quality [29]. Two authors (AWK and AZ) assessed the quality of each study using the aforementioned quality assessment scale. The quality assessment scale includes methodological quality, sample selection, sample size, outcome measurement, and statistical analysis used. In cases of disagreement between the two authors the third author (SS) is involved in the resolution.

## Data collection process

Data were extracted using the Joanna Briggs institute data extraction form for observational studies [30]. Firstly, identified articles were imported to EndNote X6 to identify and remove duplicated articles. Secondly, important data were extracted using the prepared data extraction format independently by two authors (AZ and AWK). For the first three outcomes of this systematic review and meta-analysis (knowledge, attitude and practice) the extracted data includes the primary author name, study year, publication status, publication month, publication year, study design, sample size, prevalence, study region, determinant factors, and quality of the study. Data were extracted using 2 by 2 tables for the second objective of this systematic review and meta-analysis (determinants of antenatal physical exercise practice among pregnant women). All data extraction activity was done by two authors (AWK and AZ). Finally, data analysis was done by STATA software version 17 [31].

## Outcome measurement and data items

The level of knowledge, attitude and practice towards antenatal physical exercise and factors associated with practices of antenatal physical exercise among pregnant women were the outcome variables of this systematic review and meta-analysis.

The practice of antenatal physical exercise was measured in accordance with the recommendation of ACOG in which woman who has engaged in physical exercise at least three times a week for a minimum of 20 minutes were considered to have adequate practice of antenatal physical exercise otherwise they were categorized as having inadequate practices.

Knowledge of pregnant women towards antenatal physical exercise was assessed by soliciting them about the benefits and contraindications of physical exercise in which the correct responses were coded as "1" and incorrect responses were coded as "0". Woman who scored the mean value or above from the knowledge assessment items were categorized as having adequate knowledge whereas those women scored below the mean value were categorized as having inadequate knowledge towards antenatal physical exercise. Attitude of woman regarding antenatal physical exercise were measured using composite of items. Favorable responses were coded as "1" and unfavorable responses were coded as "0". Women who scored the mean value or above from attitude assessment items were categorized as having favorable attitude towards antenatal physical exercise and those who scored below the mean value were categorized as having unfavorable attitude towards antenatal physical exercise.

## Effect measures

Descriptive statistics including proportions and frequencies were used to present outcome variables including knowledge, attitude and practice of pregnant women towards antenatal physical exercise. In addition, this systematic review and meta-analysis used odds ratio to measure the effects of exposure variables on the outcome variable.

## Syntheses methods and reporting bias assessment

Data were extracted using the Microsoft Excel spreadsheet format and imported to STATA software version 17 for analysis. Heterogeneity among studies was checked using the Cochrane Q test and $I^2$ statistics. The level of heterogeneity among studies is quantified by $I^2$ statistics. Accordingly, if the result of $I^2$ is 0% to 40% it is mild heterogeneity, 30 to 60% would be moderate heterogeneity, 50 to 90% would be substantial heterogeneity; and 75 to 100% would be considerable heterogeneity [32]. The weighted Inverse variance random effect model was used to estimate the pooled level of knowledge, attitude and practice towards antenatal physical exercise in Ethiopia. The random effect model was used due to the observed considerable heterogeneity ($I^2$ = 98.4%) among studies. Forest plot was used to illustrate the pooled level of knowledge, attitude and practices of antenatal exercise among pregnant women with 95% CI. Publication bias was checked visually using a funnel plot and statistically using Egger's regression test, with P<0.05 indicating significant publication bias. Sensitivity analysis was done to estimate the effect of a single study on the overall estimate of the level of knowledge, attitude and practices of antenatal physical exercise among pregnant women and sub-group analysis was done using regions where the studies are conducted.

## Certainty assessment

Grading of Recommendations Assessment, Development and Evaluation (GRADE) assessment was used to assess the overall certainty of the evidence. A GRADE assessment comprises risk of bias to the internal validity of results, consistency of results across studies, directness and precision of results, and likelihood of publication bias. The overall quality of evidence is then categorized as high, moderate, low or very low [33]. Grading of Recommendations Assessment, Development and Evaluation assessments were conducted for the primary outcome included in the meta-analysis. Two independent researchers (AWK and AZ) performed the GRADE assessments.

# Results

## Study selection

A total of 311 studies were retrieved in our literature search on knowledge, attitude, practice and associated factors towards antenatal physical exercise among pregnant women in Ethiopia. Of these, 110 articles were identified as duplicates by endnote software and removed, and 166 additional articles were removed by reading the titles and abstracts. The remaining 35 articles were eligible for assessment of the full length article. From the 35 eligible articles for retrieval, we retrieved 31 full length articles; the remaining four articles were excluded because of unable to access the full length article. After careful assessment of retrieved articles, 20 articles were excluded due to reasons mentioned in **Fig 1** below. Accordingly, 11 studies have fulfilled the inclusion criteria and incorporated in this systematic review and meta-analysis.

## Characteristics of included studies

The included studies are three in Tigray region [21,25,34], while Amhara region [18,23], SNNPR (Southern Nations Nationalities and People's Region) region [20,24] and Addis Ababa city contributed two studies each [35,36], and Harar [37] and Sidama regions represented by one study each [19]. All of the included articles were cross-sectional studies [18–21,23–25,34–37], and the included sample ranges from 240 to 806 pregnant women. The characteristics of the included studies are summarized in **Table 1** below.

## Results of syntheses and reporting bias

**Knowledge of pregnant women towards antenatal physical exercise.** The overall prevalence of good knowledge towards antenatal physical exercise among pregnant women in Ethiopia was 46.04% (95% CI; 44.45%-47.63%). Substantial level of heterogeneity was observed from the estimate of knowledge of antenatal exercise among pregnant women as demonstrated by $I^2$ statistics ($I^2$ = greater than 96% for three of the variables, p<0.001). The finding was summarized using a forest plot presented in **Fig 2** representing knowledge of antenatal physical exercise among pregnant women in Ethiopia. Due to the observed heterogeneity, sub-group analysis was done using administrative regions where studies were conducted.

**Attitude of women towards antenatal physical exercise.** The overall prevalence of favorable attitude towards antenatal physical exercise among pregnant women in Ethiopia was 43.71% (95% CI: 41.95%-45.46%) Substantial level of heterogeneity was observed from the estimate of attitude of antenatal exercise among pregnant women as demonstrated by $I^2$ statistics ($I^2$ = greater than 96% for three of the variables, p<0.001) **Fig 3.**

**Practices of women towards antenatal physical exercise.** The pooled prevalence of good practice towards antenatal physical exercise among pregnant women in Ethiopia was 34.06% (95% CI: 20.04%-48.08%). Substantial level of heterogeneity was observed in practices of antenatal physical exercise among pregnant women as demonstrated by $I^2$ statistics ($I^2$ = greater than 96% for three of the variables, p<0.001). The finding was summarized using a forest plot presented in **Fig 4**. Due to the observed heterogeneity, sub-group analysis was done using administrative regions where studies were conducted.

**Publication bias.** Publication bias was checked using funnel plot and egger's test. The egger's test result was found not statistically significant; hence the study has no publication bias. Besides, the funnel plot was symmetrical that corroborates the egger's statistical test result in which no publication bias was identified. The funnel plots for knowledge, attitude and practice of pregnant women towards antenatal exercise in Ethiopia were depicted in **Figs 5–7** respectively.

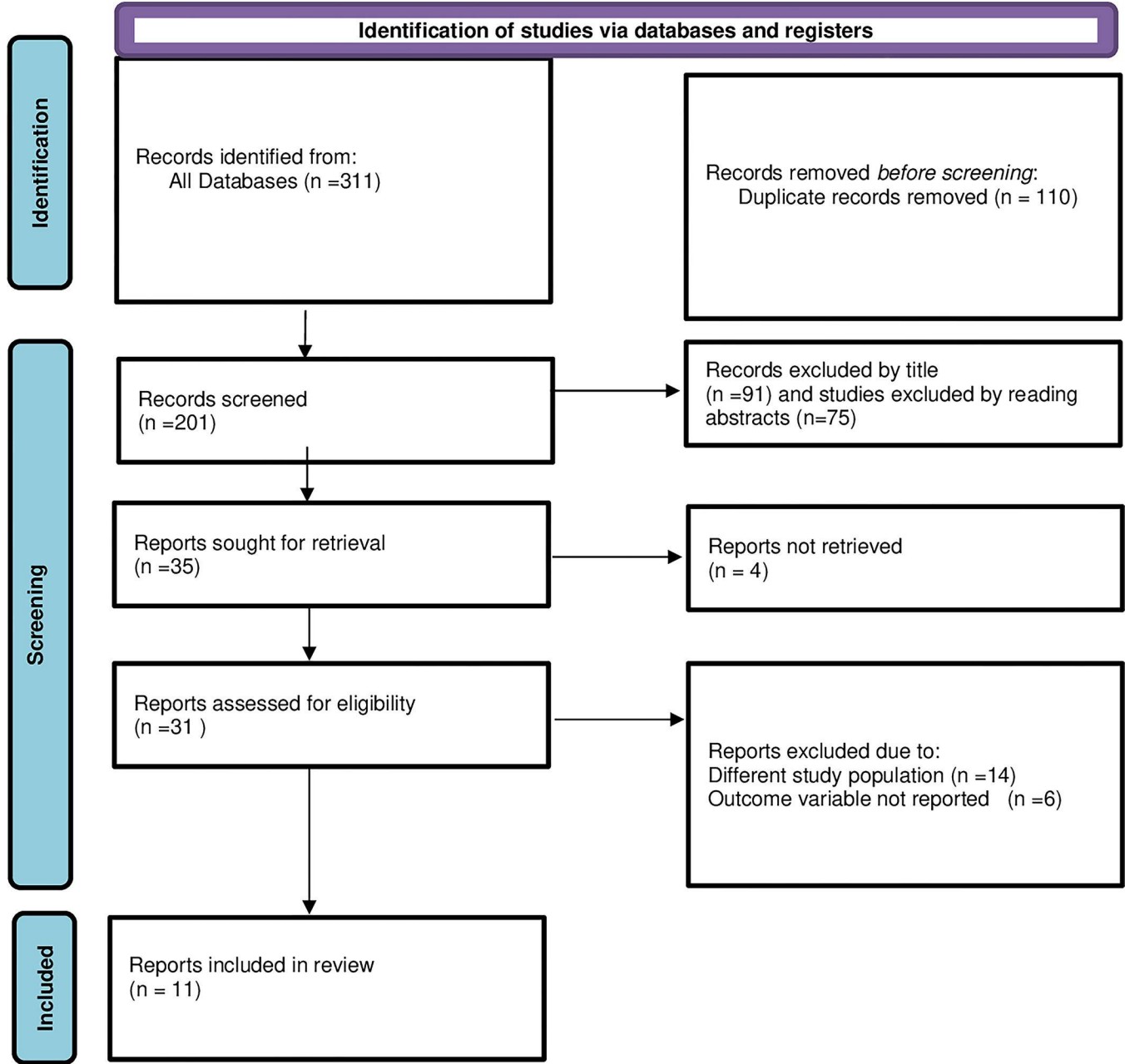

**Fig 1. Flow chart illustrating selection of studies for systematic review and meta-analysis on knowledge, attitude and practice of antenatal exercise among pregnant women in Ethiopia, 2023.**

**Sub-group analysis of knowledge attitude and practice of pregnant women towards antenatal physical exercise.** The knowledge of pregnant women towards antenatal physical exercise was lowest in Sidama region with 26.17% (95% CI: 22.65%-29.68%) level of good knowledge; whereas pregnant women in Harar region had the highest level of good knowledge towards antenatal physical exercise where three in four pregnant women had good knowledge on antenatal physical exercise (75.83% (955 CI: 70.42%-81.25%)).

Pregnant women in Tigray region had the highest level of favorable attitude with 56.08% (95% CI: 49.99%-62.17%) towards antenatal physical exercise than the rest of regions in

**Table 1. Characteristics of included studies on the systematic review and meta-analysis entitled knowledge attitude and practices of antenatal exercise among pregnant women in Ethiopia.**

| Author | Study year | Study region | Study design | Sample size | Prevalence of recommended KAP towards antenatal exercise | | | Study quality |
|---|---|---|---|---|---|---|---|---|
| | | | | | Good knowledge | Favorable attitude | Good Practice | |
| Negash S,et al [35] | 2021 | Addis Ababa | Cross-sectional | 806 | 50.37% | 27.9% | 22.3% | Good |
| Beyene MM et al [20] | 2022 | SNNPR | Cross-sectional | 422 | 46.3% | 46% | 32.9% | good |
| Sitot A et al [25] | 2020 | Tigray | Cross-sectional | 255 | 51% | 56% | 16.6% | good |
| Janakiraman B et al [23] | 2021 | Amhara | Cross-sectional | 349 | 39.5% | 55.3% | 37.9% | good |
| Beyene T et al [36] | 2018 | Addis Ababa | Cross-sectional | 355 | 43.4% | 52.1% | 24.8% | good |
| Gebregziabher D et al [34] | 2019 | Tigray | Cross-sectional | 442 | 78.05% | Not measured | Not measured | good |
| Legesse M et al [24] | 2020 | SNNPR | Cohort study | 247 | 47.2% | Not measured | Not measured | good |
| Belachew DZ et al [19] | 2023 | Sidama | Cross-sectional | 606 | 43.7% | Not measured | 25.5% | good |
| Hailemariam TT et al [21] | 2020 | Tigray | Cross-sectional | 299 | Not measured | Not measured | 48.5% | good |
| Ergicho SE et al [37] | 2017 | Harar | Cross-sectional | 240 | 75.8% | 50.8% | Not measured | good |
| Bayisa D et al [18] | 2020 | Amhara | Cross-sectional | 475 | 55.8% | 53.3% | Not measured | good |

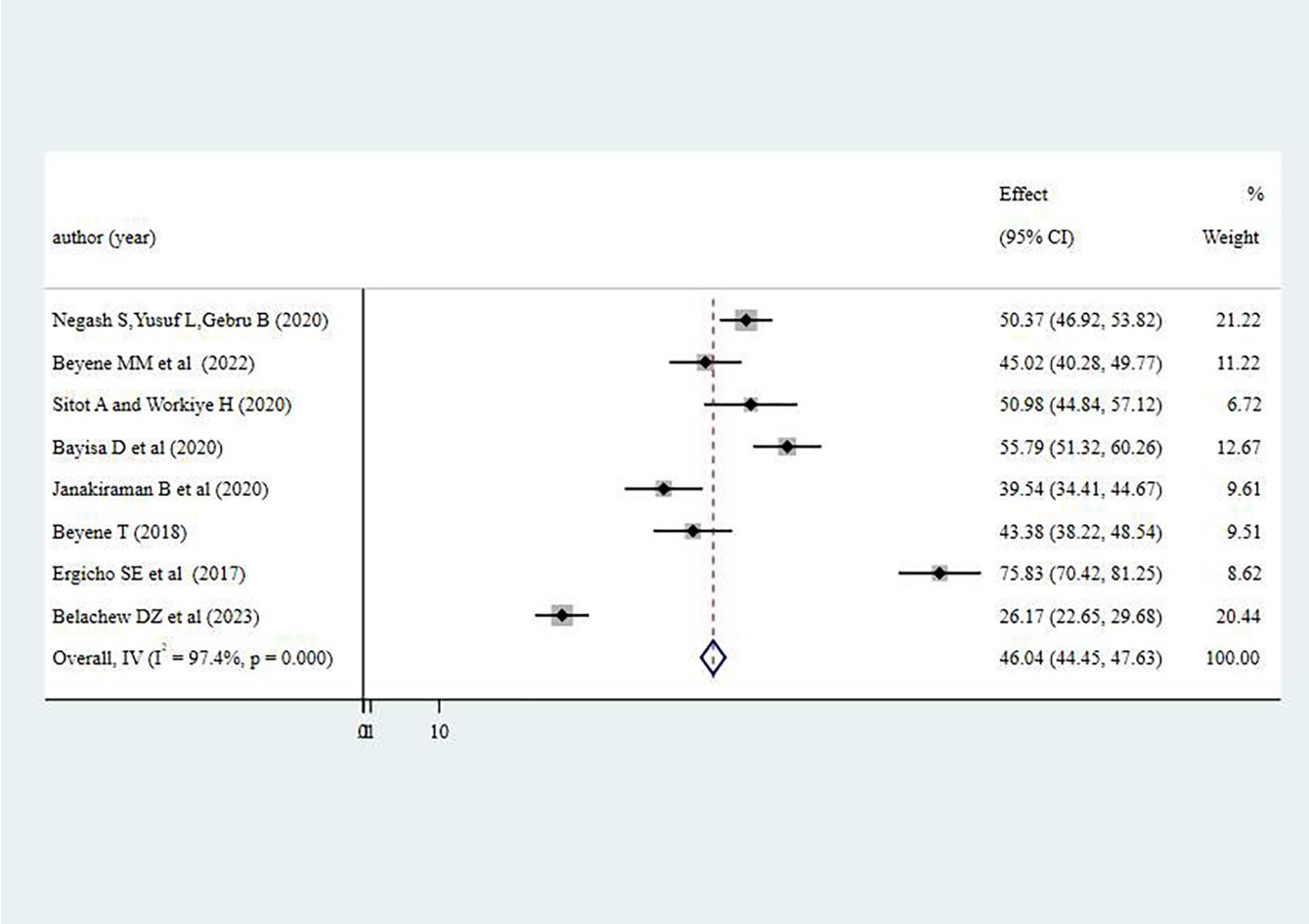

**Fig 2. Pooled level of good knowledge on antenatal physical exercise among pregnant women in Ethiopia, 2023.**

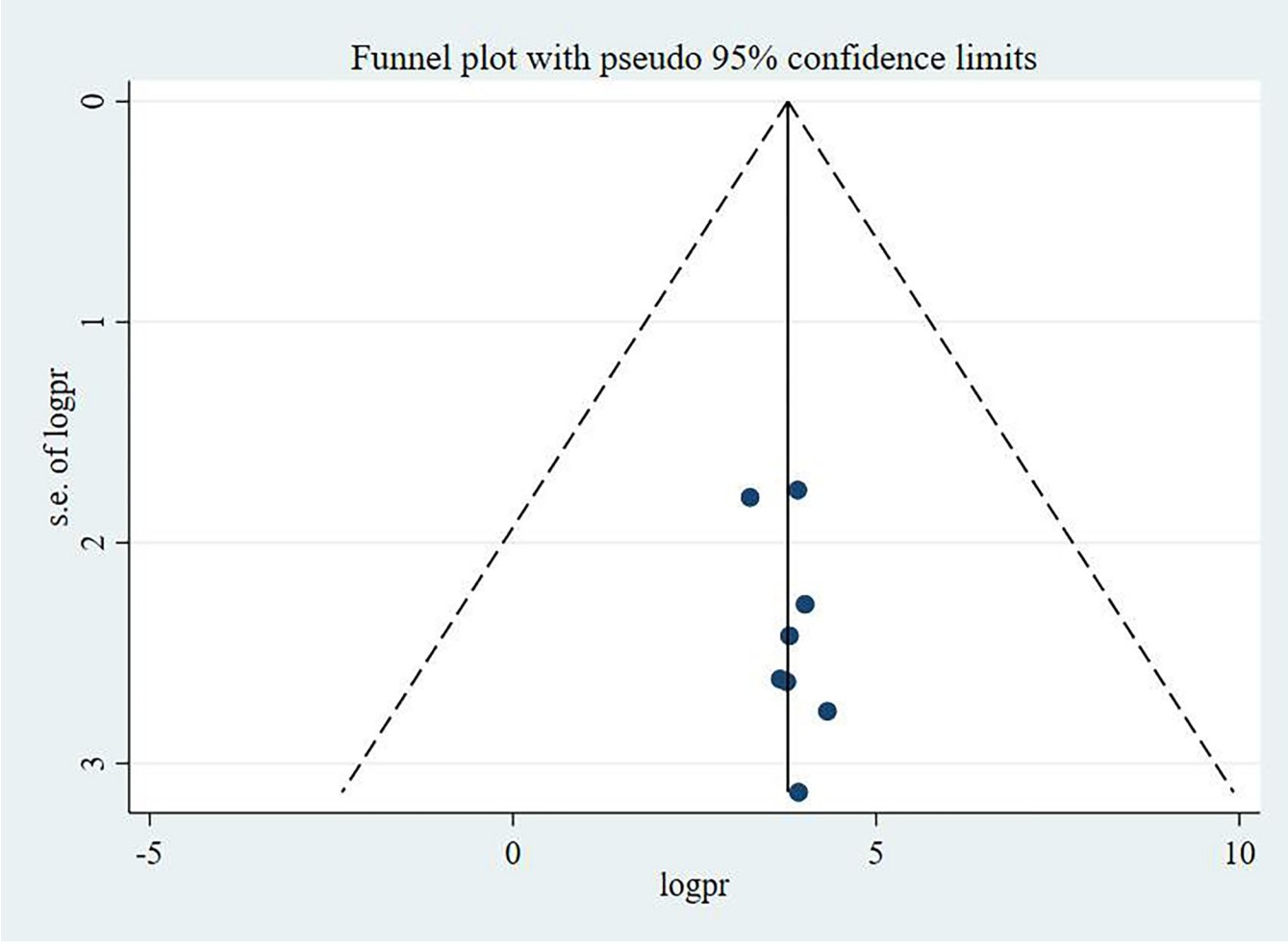

**Fig 3. Funnel plot showing the symmetric distribution of articles on pregnant women's knowledge on antenatal physical exercise in Ethiopia, 2023.**

Ethiopia, on the contrary women in Addis Ababa city had the least level of favorable attitude towards antenatal physical exercise 39.92% (95%% CI: 16.21%-63.63%).

With regard to practices of women towards antenatal physical exercise, pregnant women in Tigray region had the highest level of good antenatal physical exercise practices than women in other regions of Ethiopia; on the other hand pregnant women in Addis Ababa were least in practicing antenatal physical exercises.

**Sensitivity analysis.** A random effect model finding illustrated that no single study has influenced the overall prevalence of knowledge, attitude and practice of pregnant women towards antenatal exercise in Ethiopia (**Figs 8–10** for knowledge, attitude and practice respectively).

## Factors associated with practice of antenatal physical exercise among pregnant women

Knowledge towards antenatal physical exercise, attitude towards antenatal physical exercise, pre-pregnancy habits of physical exercise, and having college diploma or above education status were factors significantly associated with practice of antenatal physical exercise.

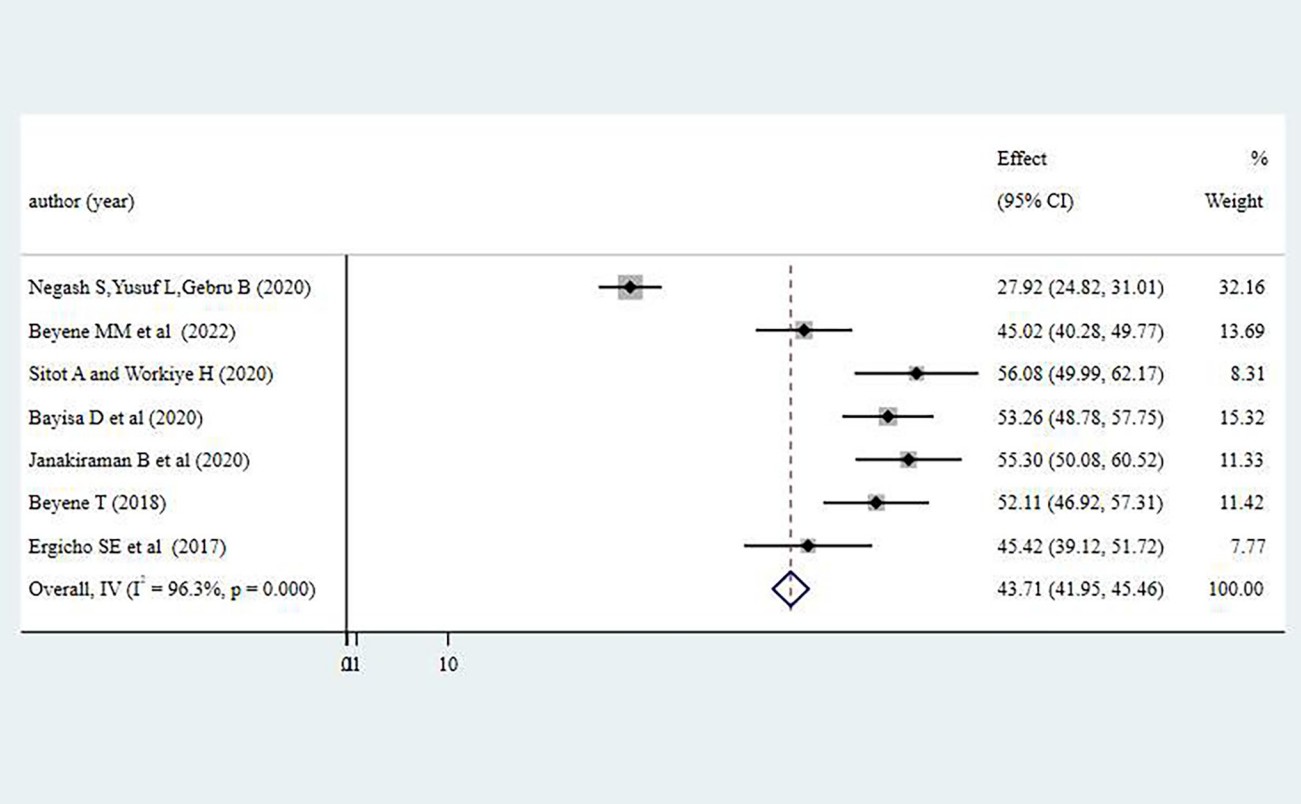

**Fig 4. Pooled level of favorable attitude on antenatal physical exercise among pregnant women in Ethiopia, 2023.**

Pregnant women with adequate knowledge on benefits and contraindications of antenatal physical exercise were 2.3 times more likely to engage in physical exercise during pregnancy compared to women who have inadequate knowledge on antenatal physical exercise with AOR 2.38 (95% CI:1.80–3.14). Women who had unfavorable attitude towards antenatal physical exercise were 57% less likely to practice antenatal physical exercise during pregnancy compared to the counter part with AOR 0.43 (95% CI: 0.32–0.58). Pre-pregnancy habit of antenatal physical exercise reduced the likelihood of antenatal physical exercise by 56% among pregnant women with AOR 0.44 (95% CI: 0.24–0.79). Pregnant women who had college diploma or above level of education status were 3.39 times more likely to practice antenatal physical exercise compared to illiterate women with AOR 3.39 (95% CI: 1.92–5.98) (**Table 2**).

## Assessment of certainty of evidence

As all of the included studies were cross sectional, we stood at low certainty of evidence and further assessed certainty of the evidence using the five down grading and the three up grading elements of the GRADE certainty assessment criteria [33]. Accordingly, the generated

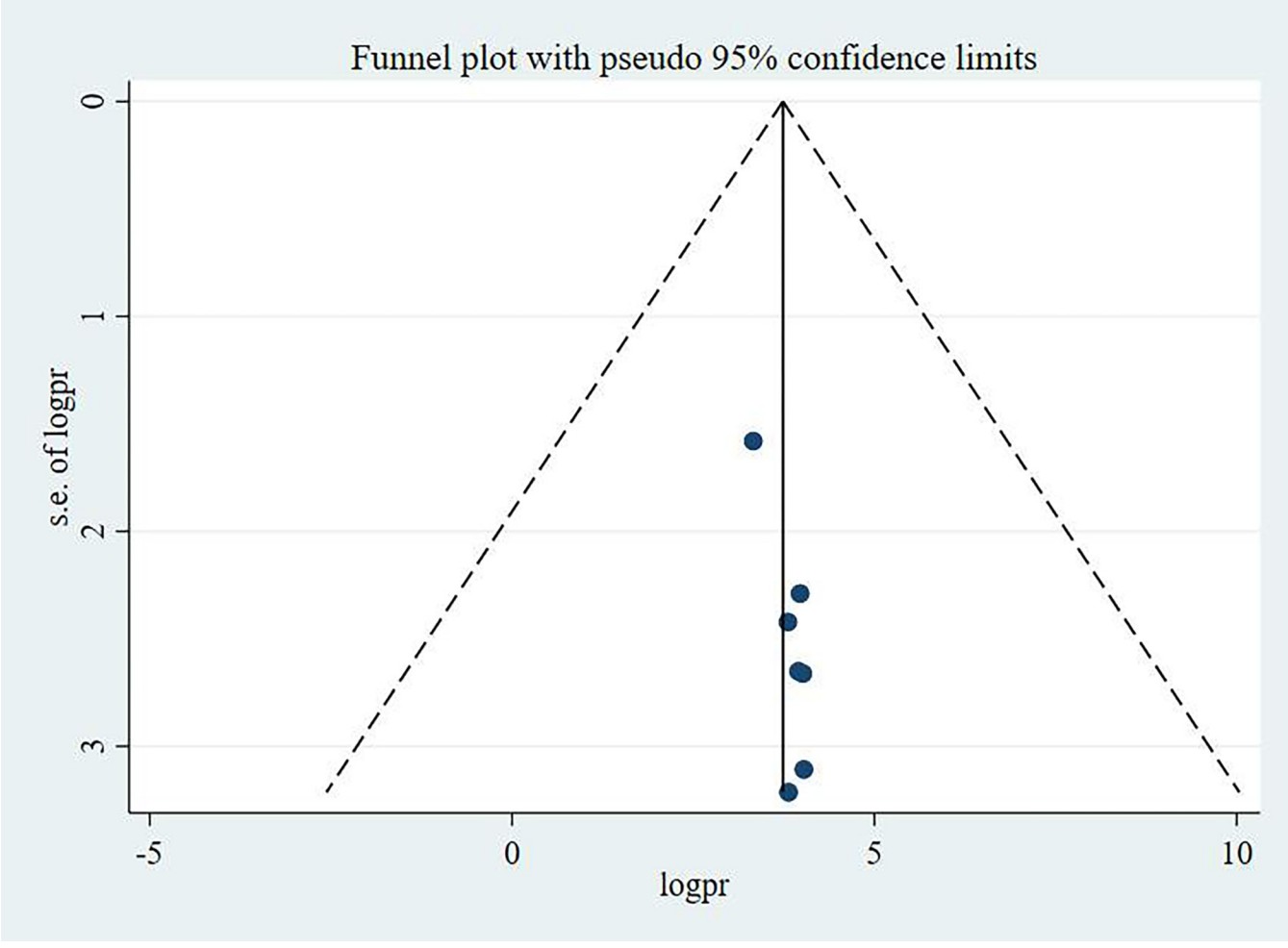

**Fig 5. Funnel plot showing the symmetric distribution of articles on pregnant women's attitude on antenatal physical exercise in Ethiopia, 2023.**

evidences tend to suffer from inconsistencies due to the fact that few studies were included in the review. It was demonstrated by significantly identified heterogeneity of estimate of the outcome variable across different administrative regions of Ethiopia. However, there was no identified effect of publication bias and risk of bias as all included studies have followed robust methods and egger's test and funnel plots demonstrated the absence of publication bias. There was no evidence of imprecision of the measurements of the outcome variable as the sample size was well enough for all included studies and the confidence interval of the estimated pooled prevalence was narrow. On top of that, the outcome variable was directly measured from all the included studies, no included study had indirectly measured the outcome variable of this systematic review and meta-analysis. The confounding variables were controlled by considering all the factors that may affect measurements of the estimates of variable and the estimate of the outcome variable was large enough in all included studies.

In general, the current level of knowledge, attitude and practices of pregnant women towards antenatal physical exercise in Ethiopia is highly likely to change if more studies are included; hence it warrants further studies to produce more accurate evidence that we can rely on for decision making purpose (S2 Table).

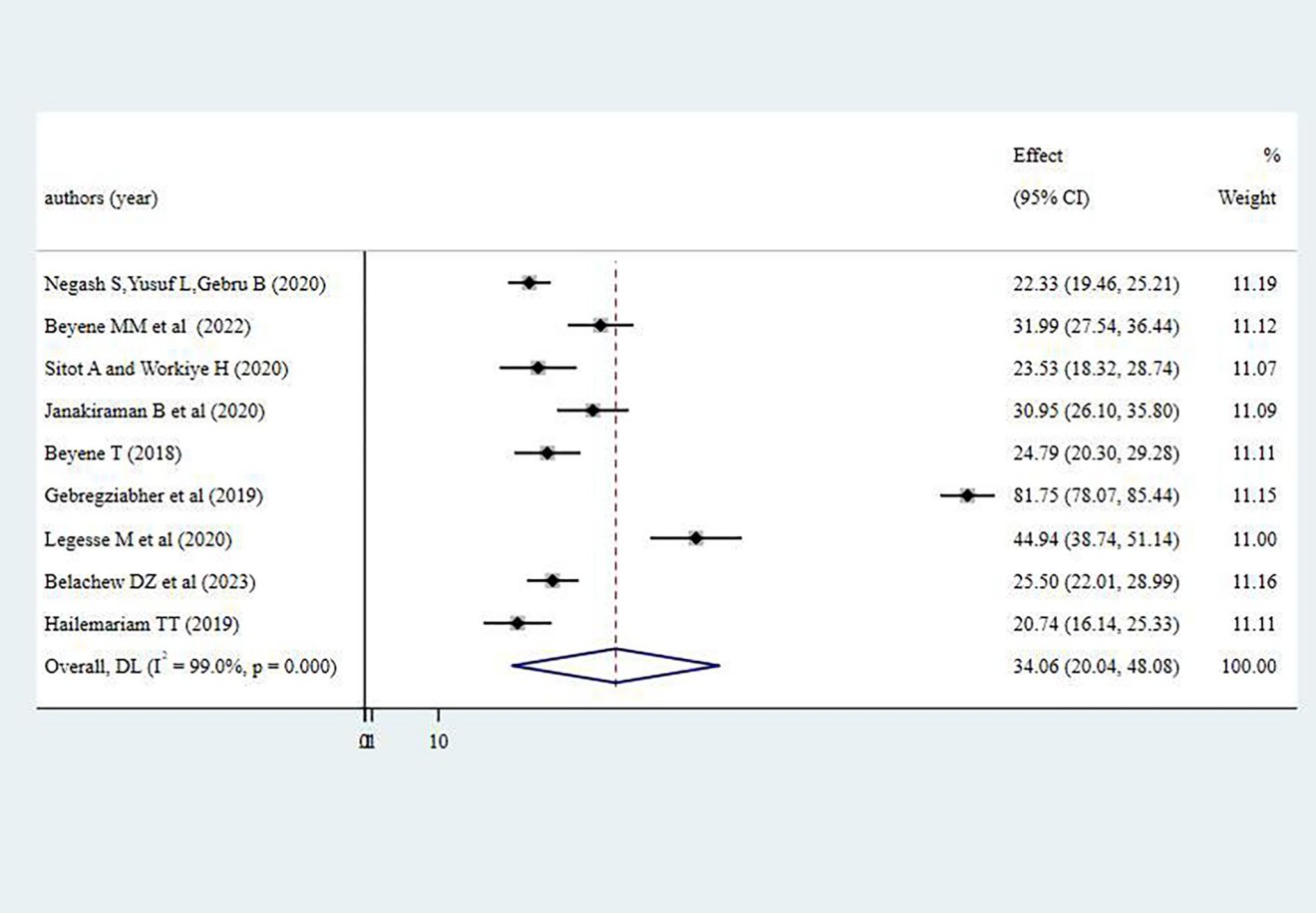

**Fig 6. Pooled level of adequate practice of antenatal physical exercise among pregnant women Ethiopia, 2023.**

## Discussion

This systematic review and meta-analysis tried to uncover the level of knowledge, attitude and practice towards antenatal physical exercise among pregnant women in Ethiopia. Accordingly, the overall level of good knowledge, favorable attitude and good practice on antenatal physical exercise is found 46.04%, 43.71% and 34.06% respectively. Though a little lower than half of pregnant women have good knowledge and favorable attitude towards antenatal physical exercise; only a third of them have engaged in a good level of antenatal physical exercise practices.

The level of good knowledge on antenatal physical exercise in this study is similar with a finding in Nigeria [38], but it is lower than the level of expectant mothers' knowledge on antenatal physical exercise in Saudi Arabia, Brazil, Sri Lanka and Ghana [39–42]. The reason might partly attributable to the fact that the level of knowledge in the case of this study is computed from many studies conducted on different sub-national settings including the rural areas of Ethiopia where the level of knowledge is expected to be low, whereas the finding from Saudi Arabia, Brazil and Ghana are from a single study in each country predominantly from urban and semi-urban areas where women health literacy is better than rural areas.

However, the level of good knowledge on antenatal physical exercise in this study is well higher than findings from two sub-national settings of Pakistan and Sri Lanka where only

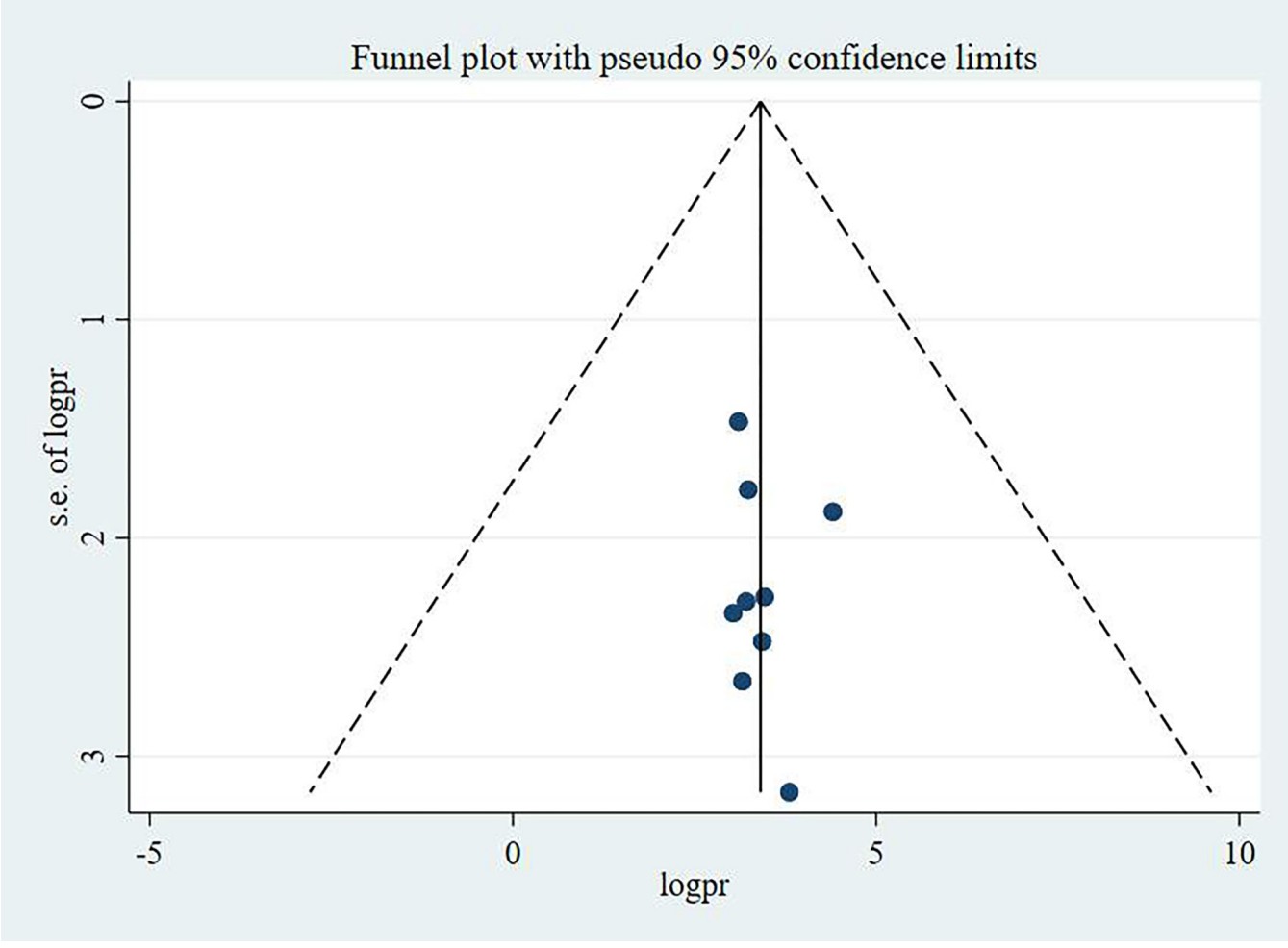

**Fig 7. Funnel plot showing the symmetric distribution of articles on pregnant women's practice on antenatal physical exercise in Ethiopia, 2023.**

20%, 17% and 35.5% of pregnant women were knowledgeable on the benefits and contraindications of antenatal physical exercise. The articles from Pakistan claimed that the presence of deeply prevailed myths regarding antenatal physical activity and exercise is responsible for the low score of knowledge on antenatal physical exercise in Pakistan [42–44].

With regard to attitude of pregnant women on antenatal physical exercise, in this study about 43.7% of pregnant women are in favor of being active during pregnancy which is far below the attitude of pregnant women in Brazil in which more than 90% of pregnant women had positive opinion for doing physical exercise during pregnancy [40]. The reason for the disparity might be due to the study in Brazil is among urban women where their level of health literacy and understanding on the essence of physical exercise during pregnancy is expected to be better than the present study where both urban and rural resident women were participated. Similarly this finding is lower than findings in India, and Pakistan [45,46]. However, women in Ethiopia have more favorable attitude towards antenatal exercise compared to women in Sri Lanka [42].

As it is already mentioned in the first paragraph of the discussion section, one in three pregnant women in Ethiopia have good antenatal physical exercise practices. This finding is higher than the level of antenatal physical exercise involvement by expectant mothers of Brazil where

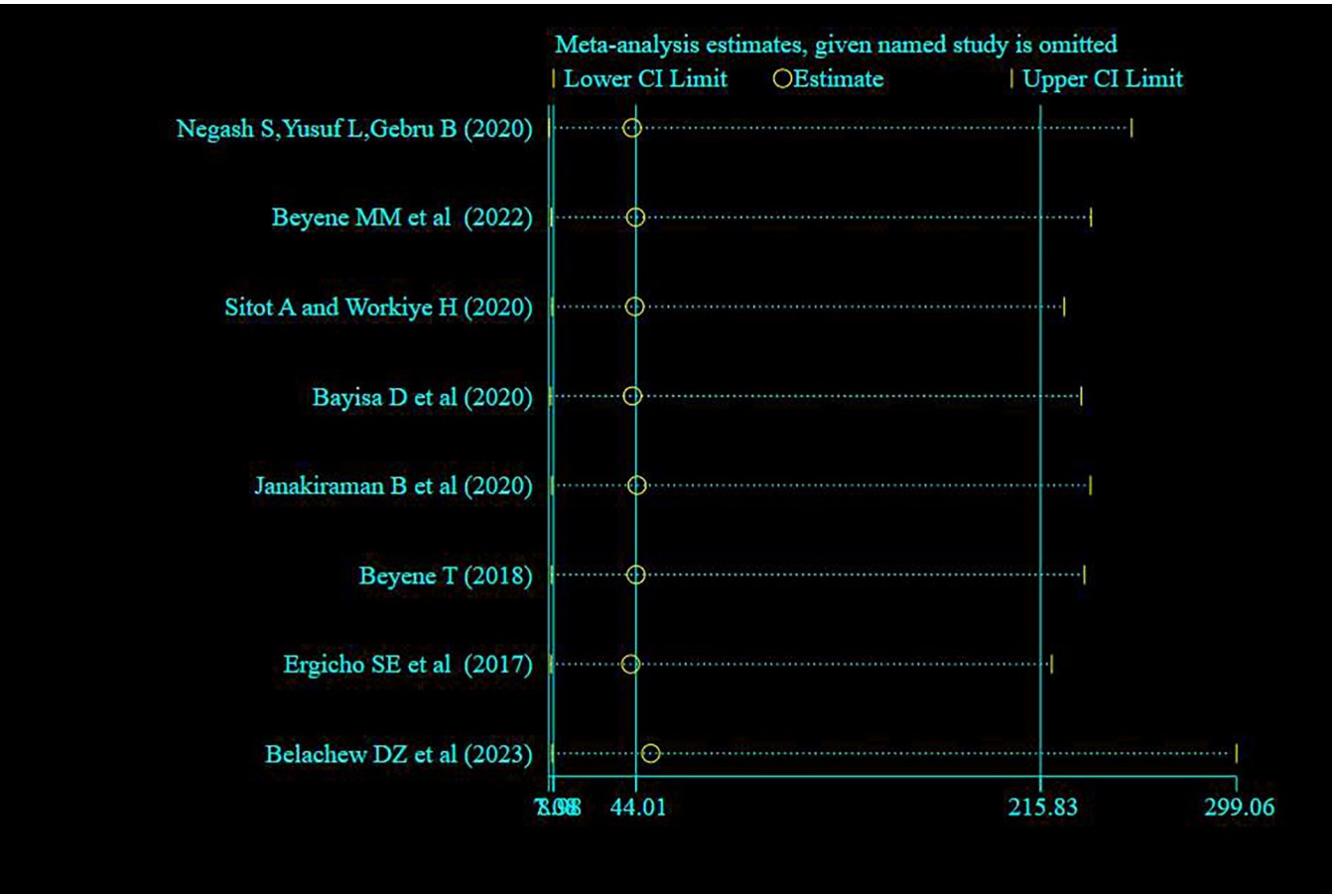

**Fig 8. Sensitivity analysis of studies included for systematic review and meta -analysis pregnant women's knowledge on antenatal physical exercise in Ethiopia, 2023.**

only one in five pregnant women was involved in antenatal physical exercise [47]. Similarly, the finding of the present study portrays higher level of antenatal physical exercise practice among pregnant women compared to the finding in Ireland, Sri Lanka, Hong Kong and Nigeria where only 21.5%, 13.6%, 23% and 10% of pregnant women were involved in moderate intensity physical exercise respectively [42,48–50]. Myriads of reasons might be responsible for the observed difference in practice of antenatal exercise including variations in socio-cultural fabrics of the study population; communities have their custom and ritual practices during pregnancy that can encourage or discourage antenatal physical exercise, variations in measurement of the outcome variable; for instance in case of the article from Ireland the study design was cohort and more objective measurement approaches of physical exercise was employed which may underestimate the level of physical activity compared to this study.

Knowledge of pregnant women towards antenatal physical exercise has significantly influenced the practice of moderate intensity physical exercise during pregnancy. It is obvious that women who are well aware of the benefits and contraindications of antenatal physical exercise are more likely to adhere to the recommended level of antenatal physical exercise. Similar finding is reported in Brazil, Saudi Arabia and Nigeria where knowledge is found as significant predictor of being active during pregnancy [39,40,50].

Besides the knowledge of pregnant women, attitude of pregnant women towards antenatal physical exercise is found a significant predictor of involvement in the recommended practice

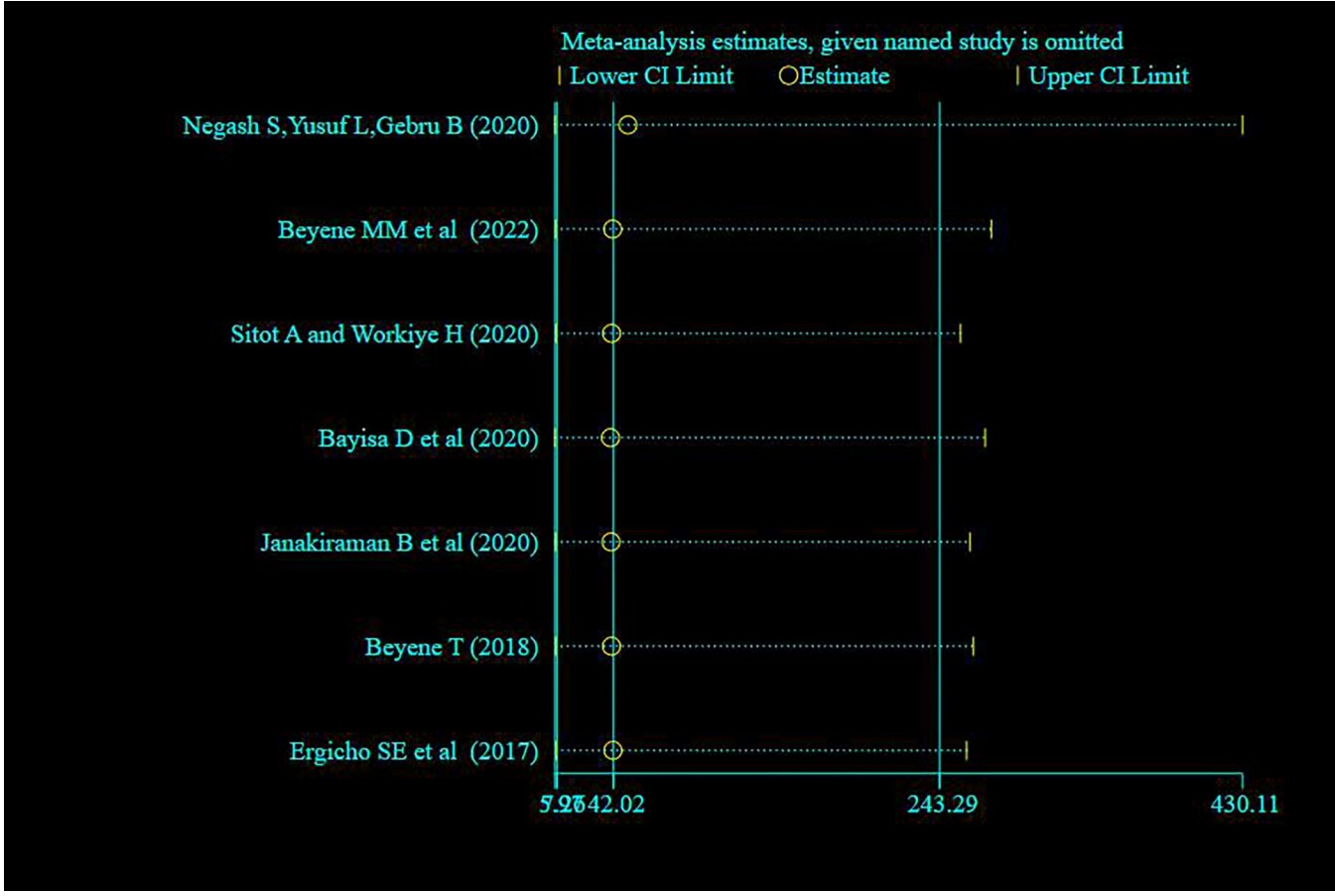

**Fig 9. Sensitivity analysis of studies included for systematic review and meta -analysis pregnant women's attitude on antenatal physical exercise in Ethiopia, 2023.**

of antenatal physical exercise. Women who had unfavorable attitude towards practicing antenatal physical exercise are more likely to be inactive during pregnancy compared to those who have favorable attitude. Other study in Nigeria have reported similar finding in which women with unfavorable attitude opt sedentary way of life during pregnancy compared to women who have favorable attitude towards antenatal physical exercise [50].

Pregnant women with college diploma and above are more likely to have good antenatal exercise practices compared to women with lower education status. This is due to the fact that women with higher education status are more likely to be well informed on the benefits of antenatal exercises and they tend to have more access to media. As a result myths and misconceptions regarding antenatal physical exercise is not common among women with higher education level. Studies in China Hong Kong, Brazil, Pakistan, Brazil Sao Paulo and Saudi Arabia have portrayed similar positive pattern of association between antenatal physical exercise and higher education level [40,43,47,49,51].

Women who had pre-pregnancy habit of physical exercise are more likely to have the recommended level of antenatal physical exercise practice. These women are cognizant of the benefits of physical exercise and tend to retain their pre-pregnancy behaviors in the antenatal periods. The studies in Brazil, Ghana and Netherlands have shown similar association of previous habit of physical exercise has positive effect for antenatal physical exercise practice during antenatal period [41,47,52].

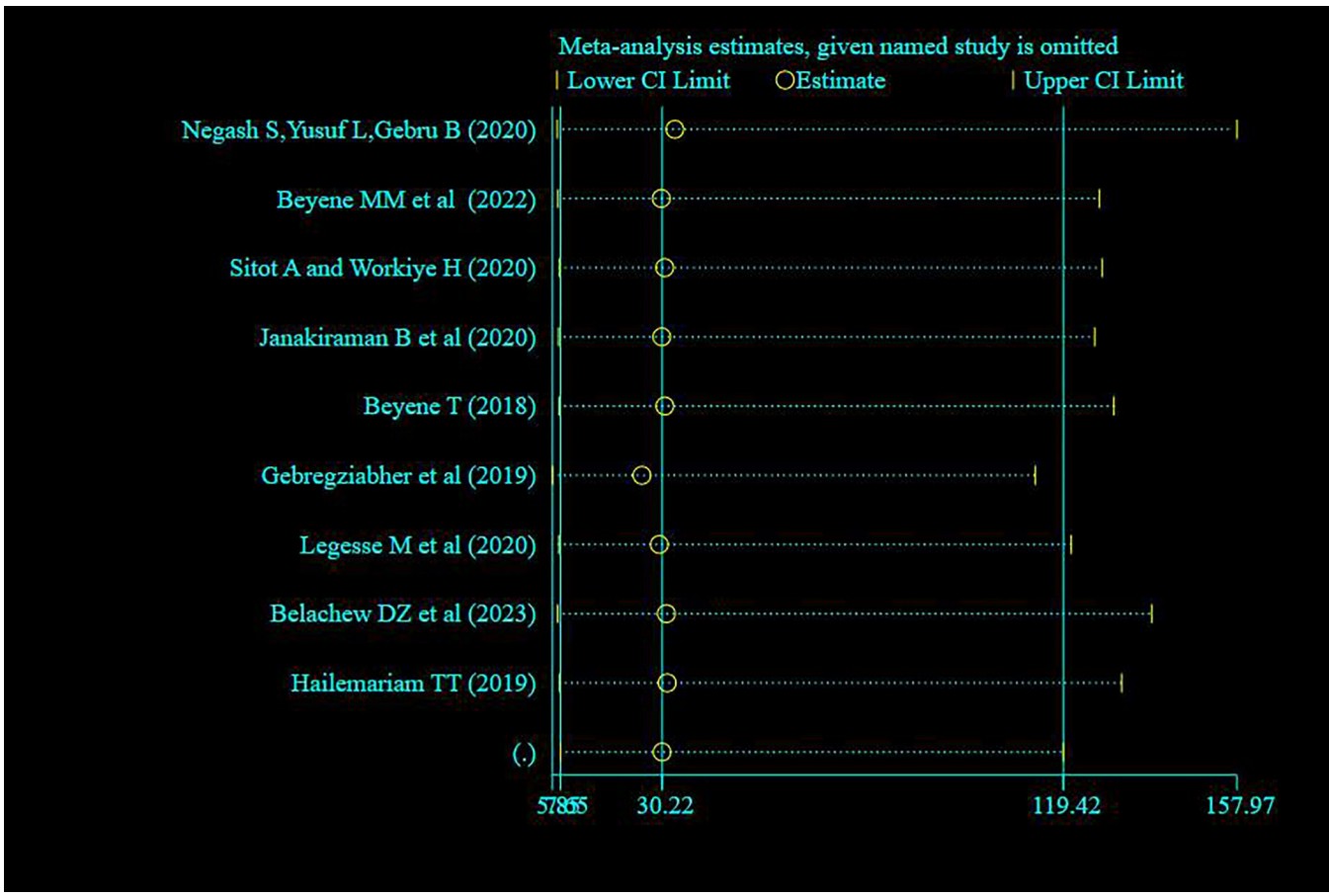

**Fig 10. Sensitivity analysis of studies included for systematic review and meta -analysis pregnant women's practice on antenatal physical exercise in Ethiopia, 2023.**

This systematic review and meta-analysis has several shortcomings that have to be considered in utilizing the findings. The first limitation is originated from the included studies; the result of this systematic review and meta-analysis is based on the already available evidence from previous studies. Consequently, the limitation and quality of the included studies do have its impact on the findings of this study. Secondly, articles published in languages other

**Table 2. Factors associated with practice of antenatal physical exercise among pregnant women in Ethiopia, 2023.**

| Variables | Authors | AOR | 95% CI | Pooled OR | 95% CI of pooled OR |
|---|---|---|---|---|---|
| Adequate knowledge on antenatal physical exercise | Negash S,et al | 2.2 | 2.14–5.00 | 2.38 | 1.80–3.14 |
| | Janakiraman B et al | 4.53 | 1.64–15.3 | | |
| | Beyene T et al | 3.76 | 1.38–8.45 | | |
| | Belachew DZ | 2.12 | 1.49–3.51 | | |
| Unfavorable attitude towards antenatal physical exercise | Negash S,et al | 0.57 | 0.38–0.86 | 0.43 | 0.32–0.58 |
| | Beyene MM et al | 0.3 | 0.2–0.5 | | |
| Lack of Experience in pre-pregnancy physical exercise | Beyene T et al | 0.42 | 0.21–0.84 | 0.44 | 0.24–0.79 |
| | Janakiraman B et al | 0.49 | 0.09–0.88 | | |
| Having College and above education status | Janakiraman B et al | 4.01 | 2.3–11.4 | 3.39 | 1.92–5.98 |
| | Beyene T et al | 2.85 | 1.68–8.45 | | |

than English language were not included in this systematic review and meta-analysis; this will introduce bias if there were articles published in other languages. Furthermore, the included studies to generate overall estimates of the outcome variables are few; as a result it might not show the true estimate of the outcome variable. Finally, only quantitative studies were considered for this systematic review and meta-analysis; it would have been more informative if both quantitative and qualitative studies had been considered.

## Conclusion

The level of knowledge, attitude and practice towards antenatal physical exercise among pregnant women is far below the recommended level in Ethiopia. More than half of pregnant women have inadequate knowledge and unfavorable attitude towards antenatal physical exercise. In spite of the proven advantages of antenatal physical exercise for both the mother and the fetus; only one third of pregnant women are reportedly active during antenatal period in Ethiopia. Pregnant women with good knowledge, favorable attitude, college level or above education status and good experience in pre-pregnancy habit of physical exercise e were significantly associated factors with antenatal physical exercise practice. It is highly essential to disseminate health information on the benefits and contraindications of antenatal physical exercise for all pregnant women during antenatal care contacts and other platforms. Myths and misconceptions that led women to have unfavorable attitude has to be refuted via integrated health information dissemination in static and outreach health service provisions. Reconfiguring the routine antenatal care guideline with a considerable session for counseling on antenatal physical exercise might help to enhance the existing level of practice. Further research is imperative regarding clients perspective on antenatal physical exercise and on the level of exercise intensity suitable for maintaining healthy pregnancy and child birth normalcy.

## Supporting information

**S1 Checklist. PRISMA 2020 checklist.**
(DOCX)

**S1 Table. Quality assessment result of included studies.**
(DOCX)

**S2 Table. GRADE Certainty of evidence assessment.**
(DOCX)

**S1 File. Search terms used in databases.**
(DOCX)

## Acknowledgments

We would like to thank all authors of the primary studies which are included in this systematic review and meta-analysis.

## Author Contributions

**Conceptualization:** Abebaw Wasie Kasahun.

**Data curation:** Abebaw Wasie Kasahun.

**Formal analysis:** Abebaw Wasie Kasahun, Amare Zewdie.

**Methodology:** Abebaw Wasie Kasahun.

**Software:** Abebaw Wasie Kasahun, Amare Zewdie.

**Supervision:** Solomon Shitu, Berhanu Abebaw Mekonnen.

**Writing – original draft:** Abebaw Wasie Kasahun.

**Writing – review & editing:** Solomon Shitu, Berhanu Abebaw Mekonnen, Michael Hawlet.

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
