## [Decision Letter · Decision Letter 0]

25 Sep 2023

PONE-D-23-25557

Knowledge, attitude and practice towards antenatal physical exercise among pregnant women in Ethiopia: systematic review and Meta-analysis

PLOS ONE

Dear Dr. Kasahun,

Thank you for submitting your manuscript to PLOS ONE. After careful consideration, we feel that it has merit but does not fully meet PLOS ONE’s publication criteria as it currently stands. Therefore, we invite you to submit a revised version of the manuscript that addresses the points raised during the review process.

We look forward to receiving your revised manuscript.

Kind regards,

Raquel Leirós-Rodríguez

Academic Editor

PLOS ONE

Journal Requirements:

https://journals.sagepub.com/doi/10.1177/1559827608320134

https://journals.lww.com/greenjournal/FullText/2020/04000/Physical_Activity_and_Exercise_During_Pregnancy.61.aspx

In your revision ensure you cite all your sources (including your own works), and quote or rephrase any duplicated text outside the methods section. Further consideration is dependent on these concerns being addressed.

Additional Editor Comments:

Dear Authors,

First of all, congratulations on your work. Both the reviewers and I consider that the research is interesting, conveys new information and of relative impact for the expert community and, therefore, should be considered for publication in this Journal.

However, it has formal errors and methodological limitations that should be addressed before possible publication in this Journal.

Kind regards

Reviewers' comments:

Reviewer's Responses to Questions

**Comments to the Author**

1. Is the manuscript technically sound, and do the data support the conclusions?

Reviewer #1: Yes

Reviewer #2: Yes

2. Has the statistical analysis been performed appropriately and rigorously? 

Reviewer #1: Yes

Reviewer #2: Yes

3. Have the authors made all data underlying the findings in their manuscript fully available?

Reviewer #1: Yes

Reviewer #2: Yes

4. Is the manuscript presented in an intelligible fashion and written in standard English?

Reviewer #1: Yes

Reviewer #2: Yes

5. Review Comments to the Author

Reviewer #1: I would like to congratulate the author on conducting a systematic review and meta-analysis on an important topic, which is so technically sound in terms of the analysis conducted.

A comment on this manuscript "Writing language needs to be improved".

Reviewer #2: The article has a very interesting idea and topic. All the necessary parts are well-written, and the analyses are done and reported well. I enjoyed reading it. Below are some areas that need to be considered for improvement, and revisions should be made.

1. The objective stated in the abstract does not align with the objective stated at the end of the introduction and the title of the study. In the abstract, it mentions associated factors to awareness and so on, while the main objective is to determine the level.

2. In the abstract, it is better to use "pooled level" or "pooled proportion" instead of "pooled prevalence" to accurately describe the findings.

3. In the section regarding the databases searched, it is advisable to remove Google because the use of this database is not customary in scientific texts

4. Quality assessment section should be written in detail with a description of the method, such as the number of assessors and how disagreements were resolved, etc

5. The first paragraph of the findings should be divided into two subheadings: search results and the characteristics of studies, and the explanations for each part should come under the same part.

6. Font plots related to the variables of knowledge, attitude, and practice will be designed in a single figure and labeled as 'Figure a' for knowledge, 'Figure b' for attitude, and 'Figure c' for practice. Additionally, sensitivity analysis plots will be generated.

6. PLOS authors have the option to publish the peer review history of their article (what does this mean?). If published, this will include your full peer review and any attached files.

Reviewer #1: No

Reviewer #2: **Yes: **Vahid Rahmanian

---

## [Author Response · Author response to Decision Letter 0]

4 Oct 2023

Response to the editor and reviewers

On behalf of all authors of the submitted manuscript, I would like to express my gratitude for your constructive comments to further enrich our work. We have made a thorough revision in accordance with the given comments. Kindly find the below mentioned point by point response/reflections or actions taken to forwarded comments and suggestions.

R: Please ensure that your manuscript meets PLOS ONE's style requirements, including those for file naming. 

A: we have revised the manuscript in accordance with PLOS One’s style requirements. Figures and supporting files name have changed to harmonize it with the journal’s requirement.

R: We noticed you have some minor occurrence of overlapping text with the following previous publication(s), which needs to be addressed:

A: Thank you, we have synthesized and rephrased some overlapping texts to make our content unique. References are corrected and appropriate references that were missed on the previous version are cited on the revised version.

R: Please include captions for your Supporting Information files at the end of your manuscript, and update any in-text citations to match accordingly. Please see our Supporting Information guidelines for more information: http://journals.plos.org/plosone/s/supporting-information.

A: We have included captions for the supporting file on the revised version of the manuscript at the end of the manuscript under the heading “Supporting information”

R: Writing language needs to be improved

A: We have made revisions to make the manuscript readable

R: The objective stated in the abstract does not align with the objective stated at the end of the introduction and the title of the study. In the abstract, it mentions associated factors to awareness and so on, while the main objective is to determine the level.

A: we have modified the objective on the abstract section to make it consistent with the rest of the manuscript sections.

R: In the abstract, it is better to use "pooled level" or "pooled proportion" instead of "pooled prevalence" to accurately describe the findings.

A: modified accordingly

R: In the section regarding the databases searched, it is advisable to remove Google because the use of this database is not customary in scientific texts

A: correction taken

R: Quality assessment section should be written in detail with a description of the method, such as the number of assessors and how disagreements were resolved, etc

A: Thank you; we have elaborated this section a little bit. As it is already mentioned two authors were involved in the quality assessment activity and an author has involved when there was disagreement between the two assessors. The remark given by the third author was taken to declare the assessment result of the study in case of disagreement between the two assessors.

R: The first paragraph of the findings should be divided into two subheadings: search results and the characteristics of studies, and the explanations for each part should come under the same part.

A: Thanks, we have revised and presented the first paragraph of the result section in two sub-headings on the revised version.

R: Font plots related to the variables of knowledge, attitude, and practice will be designed in a single figure and labeled as 'Figure a' for knowledge, 'Figure b' for attitude, and 'Figure c' for practice. Additionally, sensitivity analysis plots will be generated.

A: Thank you, it is valid comment, but we couldn’t manage it in that way, we had tried to present the funnel plots of the three variables in single figure, however it becomes more congested and not easily readable. As a result we prefer to retain a standalone figures for each of the three variables.

---

## [Decision Letter · Decision Letter 1]

31 Oct 2023

PONE-D-23-25557R1Knowledge, attitude and practice towards antenatal physical exercise among pregnant women in Ethiopia: systematic review and Meta-analysisPLOS ONE

Dear Dr. Kasahun,

Thank you for submitting your manuscript to PLOS ONE. After careful consideration, we feel that it has merit but does not fully meet PLOS ONE’s publication criteria as it currently stands. Therefore, we invite you to submit a revised version of the manuscript that addresses the points raised during the review process.

ACADEMIC EDITOR:The manuscript contains formal errors of scientific writing that need to be corrected urgently.The Conclusions are speculative and should be rewritten with precision.The submission of a properly completed PRISMA checklist is mandatory for the next submission.Correcting the manuscript according to the PRISMA checklist (i.e. providing all the information requested and in the order indicated) is mandatory.==============================

We look forward to receiving your revised manuscript.

Kind regards,

Raquel Leirós-Rodríguez

Academic Editor

PLOS ONE

Reviewers' comments:

Reviewer's Responses to Questions

**Comments to the Author**

1. If the authors have adequately addressed your comments raised in a previous round of review and you feel that this manuscript is now acceptable for publication, you may indicate that here to bypass the “Comments to the Author” section, enter your conflict of interest statement in the “Confidential to Editor” section, and submit your "Accept" recommendation.

Reviewer #3: (No Response)

Reviewer #4: (No Response)

2. Is the manuscript technically sound, and do the data support the conclusions?

Reviewer #3: Partly

Reviewer #4: Yes

3. Has the statistical analysis been performed appropriately and rigorously? 

Reviewer #3: Yes

Reviewer #4: Yes

4. Have the authors made all data underlying the findings in their manuscript fully available?

Reviewer #3: Yes

Reviewer #4: Yes

5. Is the manuscript presented in an intelligible fashion and written in standard English?

Reviewer #3: No

Reviewer #4: Yes

6. Review Comments to the Author

Reviewer #3: I suggest an initial recommendation for publication of this paper in Plos ONE provided that you satisfactorily address all my comments, and questions and be willing to revise in further iteration, if needed. Wish you a good revision time!

General comments

( you are advised to proofread the manuscript starting at the abstract to the end )

1. You presented your reports via present tense, please use “past tense “ instead. This is a basic principle of reporting manuscripts.

2. Where P.value=0.00000…, write it as P<0.001 (please, fix this issue across the manuscript)

3. Please be consistent in using words, phrases, and statistical values across your manuscript

4. Please avoid using the term “ predictors ”! use “associated factor” instead. Because this epidemiological term is employed when studies conducted using strong models ( i.e cohort, follow-up et ) are considered. I found only one article included your synthesis conducted using a cohort study design ( refer to Table 1)

Specific comments

Title

Knowledge, attitudes, and practices towards antenatal physical exercise among pregnant

women in Ethiopia: a systematic review and Meta-analysis

- Meta-analysis should be spelled as meta-analysis

- Add “A” to the subtitle ….(correct ) A systematic review and meta-analysis

Abstract (page # 2)

Introduction: ok

Methods: ok

Results: A total of eleven studies….. Insead start with, A total of 11 studies were…….. . . NB: We should mention the absolute if frequencies are less than 10 and otherwise text if less than 10.

Conclusion: ……desirable level??…… , replace with “ the recommended level”.

….. are significant predictors of antenatal physical exercise practice. Replace with…were found to be associated with a good level of practice toward antenatal physical exercise.

Methods section

Study design and setting

• Accordingly there is no registered ……replace “is” with “was”

• Last line …..Include PRISMA statement as supplementary /additional file 1

Search strategies and source information

- Please cite ….CoCo/PEO guidelines???

- I did not find search detail in the supporting file and also you need to cite in the text

Eligibility criteria

• You considered unpublished articles/preprints for your study while they are not reputable sources of evidence as they are not peer reviewed and hence are not recommended to be included! Can you explain/resolve this issue?

• Is there any justification why you prefer to include articles published at any time until the end of our search (July12, 2023)? As this is a protracted/long period. There might be new policies, recommendations, and practices during these long period

Data processing and analysis

- Please cite.. STATA software version 17 for analysis(???).NB, software, guidelines, protocols, etc need to be cited

- What is ………pooled level of good vaccine cold chain management practice in Ethiopia???. Please remove it! I understand that you wrote the manuscript on the background of one of you’re your previous works.

Results

Characteristics of included studies

- Write SNNPR in Word as it appeared first time in the text.

- Please cite articles for each of these! The included studies are three in Tigray region (citation), while Amhara region (citation ), SNNPR region (citation), and Addis Ababa city (citation), contributed two studies each(citation), and Harar (citation), and Sidama regions (citation), represented by one study each. All of the included articles are cross-sectional studies (citation), and the included sample ranges from 240 to 806 pregnant women

- Table 1 row 1 : should it be study year or publication year?

Knowledge of pregnant women towards antenatal physical exercise

- ….....please re-write 46.04% with 95% CI (44.45%-47.63%) as ……was 46.04% (95% CI: 44.45,4.63) . would you do similar across the manuscript?

Limitation

• From the sociocultural standpoint, women's awareness, and perceptions about the domain under consideration are different among people living in a country. How do authors justify pooling the estimate of good knowledge, favorable attitude, and good practices as the studies are completely heterogeneous? Can you mention this as a limitation as it could be a source of concern for the use of the results?

• You used an article published in the English language only, please add to the limitation of the study

• Included studies were included from a few regions of your country. What do you comment on about the representativeness of the information at the national level?

Add implications for research and policy

Reviewer #4: Dear editor Thank you for assigning to review manuscript PONE-D-23-25557R1, entitled "Knowledge, attitude and practice towards antenatal physical exercise among pregnant women in Ethiopia: systematic review and Meta-analysis"

Comments

1. Please arrange key words in alphabetical orders

2. On subgroup analysis paragraph two please correct this number 4949.99%

3. On discussion part level of good knowledge comparison with different country please write level of knowledge in each country and briefly clarify justification for the difference in level of knowledge with scientific reason.

7. PLOS authors have the option to publish the peer review history of their article (what does this mean?). If published, this will include your full peer review and any attached files.

Reviewer #3: **Yes: **Beshada Zerfu Woldegeorgis (MD,MPH)

Wolaita Sodo University, College of Health Sciences and Medicine, School of Medicine

Reviewer #4: No

---

## [Author Response · Author response to Decision Letter 1]

15 Nov 2023

Response to reviewers and the editor

On behalf of all authors of the submitted manuscript, I would like to express my gratitude for your constructive comments to further enrich our work. We have made a thorough revision in accordance with the given comments. Kindly find the below mentioned point by point response/reflections or actions taken to forwarded comments and suggestions.

R: The manuscript contains formal errors of scientific writing that need to be corrected urgently.

A: thank you, We have made a thorough revision to abide with scientific writings

R: The Conclusions are speculative and should be rewritten with precision.

A: we have revised and tried to be more precise on the revised version. 

R: The submission of a properly completed PRISMA checklist is mandatory for the next submission.

Correcting the manuscript according to the PRISMA checklist (i.e. providing all the information requested and in the order indicated) is mandatory

A: we have rechecked the PRISMA checklist and we have revised the manuscript in accordance with the check list

R: You presented your reports via present tense, please use “past tense “ instead. This is a basic principle of reporting manuscripts.

A: thank you, we have revised it accordingly

R: Where P.value=0.00000…, write it as P<0.001 (please, fix this issue across the manuscript)

A: corrected accordingly

R: Please be consistent in using words, phrases, and statistical values across your manuscript

A: Thank you, we have corrected the inconsistency in terminologies that we have identified

R: Please avoid using the term “ predictors ”! use “associated factor” instead. Because this epidemiological term is employed when studies conducted using strong models ( i.e cohort, follow-up et ) are considered. I found only one article included your synthesis conducted using a cohort study design ( refer to Table 1)

A: we replaced the word predictors with associated factors

R: Meta-analysis should be spelled as meta-analysis, - Add “A” to the subtitle ….(correct ) A systematic review and meta-analysis

A: Thank you, modified accordingly

R: Abstract, A total of eleven studies….. Insead start with, A total of 11 studies were…….. . . NB: We should mention the absolute if frequencies are less than 10 and otherwise text if less than 10.

Conclusion: ……desirable level??…… , replace with “ the recommended level”.

….. are significant predictors of antenatal physical exercise practice. Replace with…were found to be associated with a good level of practice toward antenatal physical exercise.

A: Thank you, all the suggestions are relevant, we took corrections on the revised version on the aforementioned comments

R: Accordingly there is no registered ……replace “is” with “was”

A: Correction taken

R: Last line …..Include PRISMA statement as supplementary /additional file 1

A: we have included a linking statement to additional file 1.

R: Search strategies and source information, Please cite ….CoCo/PEO guidelines???

A: appropriate references is cited on the revised version

R: - I did not find search detail in the supporting file and also you need to cite in the text

A: search terms employed in each database is uploaded as a supplementary file, 

R: You considered unpublished articles/preprints for your study while they are not reputable sources of evidence as they are not peer reviewed and hence are not recommended to be included! Can you explain/resolve this issue?

A: though it is obvious that peer reviewed articles tend to have high quality evidence, it doesn’t mean that unpublished evidences are with poor quality. As long as I know both published and unpublished articles can be included in a systematic review and meta-analysis as far as they fulfill the eligibility criteria and quality assessment parameters employed in the method section. 

R: Is there any justification why you prefer to include articles published at any time until the end of our search (July12, 2023)? As this is a protracted/long period. There might be new policies, recommendations, and practices during these long periods

A: in principle it is true that changes in policy and recommendations could affect personal behaviors like antenatal physical activity. However, in our case scientific report regarding antenatal physical exercise is a recent development. All the included studies are between 2015-2023; hence we have no reason to limit our search time.

Data processing and analysis

R- Please cite.. STATA software version 17 for analysis(???).NB, software, guidelines, protocols, etc need to be cited

A: reference cited on the revised version

R- What is ………pooled level of good vaccine cold chain management practice in Ethiopia???. Please remove it! I understand that you wrote the manuscript on the background of one of you’re your previous works.

A: Thank you, apologies for such inadvertent errors; you have well understood the context in which the error has occurred, thanks indeed. I have corrected it to the concept under inquiry. 

R: Please cite articles for each of these! The included studies are three in Tigray region (citation), while Amhara region (citation ), SNNPR region (citation), and Addis Ababa city (citation), contributed two studies each(citation), and Harar (citation), and Sidama regions (citation), represented by one study each. All of the included articles are cross-sectional studies (citation), and the included sample ranges from 240 to 806 pregnant women

A; Thank you, references are cited on the revised version

R- Table 1 row 1 : should it be study year or publication year?

A; both are applicable, for published articles we took publication year whereas we can only take study year for unpublished sources. If it has to be included we can modify to (study/publication year)

R: Knowledge of pregnant women towards antenatal physical exercise

- ….....please re-write 46.04% with 95% CI (44.45%-47.63%) as ……was 46.04% (95% CI: 44.45,4.63) . would you do similar across the manuscript?

A: Corrected accordingly

R; • From the sociocultural standpoint, women's awareness, and perceptions about the domain under consideration are different among people living in a country. How do authors justify pooling the estimate of good knowledge, favorable attitude, and good practices as the studies are completely heterogeneous? Can you mention this as a limitation as it could be a source of concern for the use of the results?

A: One of the reasons for having a systematic review and meta-analysis is the existence of inconsistent level of women’s KAP towards antenatal physical exercise. Pooling will not be a problem as far as the measurements of variables are consistent. If the level of awareness is unequivocal across different settings, there will not be a rationale to compute a single estimate using a meta-analysis.

• You used an article published in the English language only, please add to the limitation of the study

A: thank you, incorporated as a limitation on the revised version of the manuscript

R: Included studies were included from a few regions of your country. What do you comment on about the representativeness of the information at the national level?

A: The included studies are few and might not fully represent all administrative regions in Ethiopia; it would have been more informative if there were additional studies to be included in this review and meta-analysis. It has already mentioned as a limitation to enable readers made careful interpretation of the results.

R: Add implications for research and policy

A: I have included its implication for policy and research on conclusion part of the revised version of the manuscript.

R: Please arrange key words in alphabetical orders

A: Thank you, it is corrected on the revised version

R: On subgroup analysis paragraph two please correct this number 4949.99%

A: thank you, the typos are corrected

R: On discussion part level of good knowledge comparison with different country please write level of knowledge in each country and briefly clarify justification for the difference in level of knowledge with scientific reason

A: we have made comparison across different settings, and actual and speculative justifications are forwarded for the observed differences on the variable under inquiry.

---

## [Decision Letter · Decision Letter 2]

20 Nov 2023

Knowledge, attitude and practice towards antenatal physical exercise among pregnant women in Ethiopia: systematic review and Meta-analysis

PONE-D-23-25557R2

Dear Dr. Kasahun,

We’re pleased to inform you that your manuscript has been judged scientifically suitable for publication and will be formally accepted for publication once it meets all outstanding technical requirements.

Kind regards,

Raquel Leirós-Rodríguez

Academic Editor

PLOS ONE

Additional Editor Comments (optional):

Dear Authors,

First of all, I would like to congratulate you on your efforts to improve this manuscript.

Taking into account the opinion of the Reviewers and my own, we all consider that the manuscript meets the formal and methodological requirements necessary to be accepted for publication.

Kind regards

Reviewers' comments:

Reviewer's Responses to Questions

**Comments to the Author**

1. If the authors have adequately addressed your comments raised in a previous round of review and you feel that this manuscript is now acceptable for publication, you may indicate that here to bypass the “Comments to the Author” section, enter your conflict of interest statement in the “Confidential to Editor” section, and submit your "Accept" recommendation.

Reviewer #3: All comments have been addressed

Reviewer #4: All comments have been addressed

2. Is the manuscript technically sound, and do the data support the conclusions?

Reviewer #3: Yes

Reviewer #4: Yes

3. Has the statistical analysis been performed appropriately and rigorously? 

Reviewer #3: Yes

Reviewer #4: Yes

4. Have the authors made all data underlying the findings in their manuscript fully available?

Reviewer #3: Yes

Reviewer #4: Yes

5. Is the manuscript presented in an intelligible fashion and written in standard English?

Reviewer #3: Yes

Reviewer #4: Yes

6. Review Comments to the Author

Reviewer #3: (No Response)

Reviewer #4: (No Response)

7. PLOS authors have the option to publish the peer review history of their article (what does this mean?). If published, this will include your full peer review and any attached files.

Reviewer #3: **Yes: **Beshada Zerfu Woldegeorgis (MD,MPH)

Wolaita Sodo University

Reviewer #4: No

---

## [Editor Report · Acceptance letter]

4 Dec 2023

PONE-D-23-25557R2 

Knowledge, attitude and practice towards antenatal physical exercise among pregnant women in Ethiopia: A systematic review and meta-analysis 

Dear Dr. Kasahun:

I'm pleased to inform you that your manuscript has been deemed suitable for publication in PLOS ONE. Congratulations! Your manuscript is now with our production department. 

Kind regards, 

on behalf of

Dr. Raquel Leirós-Rodríguez 

Academic Editor

PLOS ONE